# Summarizing Multiple Documents with Conversational Structure for Meta-Review Generation

**Miao Li**[1] and **Eduard Hovy**[1,2] and **Jey Han Lau**[1]

[1]School of Computing and Information Systems, The University of Melbourne
[2]Language Technologies Institute, Carnegie Mellon University
miao4@student.unimelb.edu.au, {eduard.hovy, laujh}@unimelb.edu.au

## Abstract

We present PEERSUM, a novel dataset for generating meta-reviews of scientific papers. The meta-reviews can be interpreted as abstractive summaries of reviews, multi-turn discussions and the paper abstract. These source documents have rich inter-document relationships with an explicit hierarchical conversational structure, cross-references and (occasionally) conflicting information. To introduce the structural inductive bias into pre-trained language models, we introduce RAMMER (Relationship-aware Multi-task Meta-review Generator), a model that uses sparse attention based on the conversational structure and a multi-task training objective that predicts metadata features (e.g., review ratings). Our experimental results show that RAMMER outperforms other strong baseline models in terms of a suite of automatic evaluation metrics. Further analyses, however, reveal that RAMMER and other models struggle to handle conflicts in source documents of PEERSUM, suggesting meta-review generation is a challenging task and a promising avenue for further research.[1]

## 1 Introduction

Text summarization systems need to recognize internal relationships among source texts and effectively aggregate and process information from them to generate high-quality summaries (El-Kassas et al., 2021). It is particularly challenging in multi-document summarization (MDS) due to the complexity of the relationships among (semi-)parallel source documents (Ma et al., 2020). However, existing MDS datasets do not provide explicit inter-document relationships among the source documents (Liu et al., 2018; Fabbri et al., 2019; Ghalandari et al., 2020; Lu et al., 2020) although inter-document relationships may also exist in nature and should be considered in methodology (Fabbri

---

[1]The dataset and code are available at https://github.com/oaimli/PeerSum

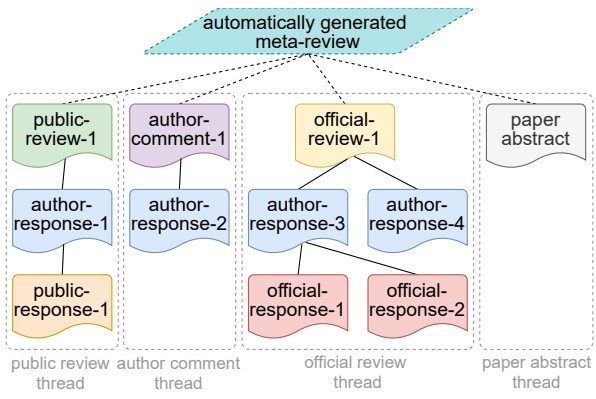

Figure 1: An illustration of the hierarchical conversational structure that PEERSUM features.

et al., 2019). This makes it hard to research inter-document relationship comprehension for information integration and aggregation in abstractive text summarization.

To enable this, we introduce PEERSUM, an MDS dataset for automatic meta-review generation. We formulate the creation of meta-reviews as an abstractive MDS task as the meta-reviewer needs to comprehend and carefully summarize information from individual reviews, multi-turn discussions between authors and reviewers and the paper abstract. From an application perspective, generating draft meta-reviews could serve to reduce the workload of meta-reviewers, as meta-reviewing is a highly time-consuming process for many scientific publication venues.

PEERSUM features a hierarchical conversational structure among the source documents which includes the reviews, responses and the paper abstract in different threads as shown in Figure 1. It has several distinct advantages over existing MDS datasets: (1) we show that the meta-reviews are largely faithful to the corresponding source documents despite being highly abstractive; (2) the source documents have rich inter-document relationships with an explicit conversational structure;

(3) the source documents occasionally feature *conflicts* which the meta-review needs to handle as reviewers may have disagreement on reviewing a scientific paper, and we explicitly provide indicators of conflict relationships along with the dataset; and (4) it has a rich set of metadata, such as review rating/confidence and paper acceptance outcome — the latter which can be used for assessing the quality of automatically generated meta-reviews. These make PEERSUM serve as a probe that allows us to understand how machines can reason, aggregate and summarise potentially conflicting opinions.

However, there is limited study on abstractive MDS methods that can recognize relationships among source documents. The most promising approaches are based on graph neural networks (Li et al., 2020, 2023), but they introduce additional trainable parameters, and it is hard to find effective ways to construct graphs to represent source documents. To make pre-trained language models have the comprehension ability of complex relationships among source documents for MDS, we propose RAMMER, which uses *relationship-aware attention manipulation* — a lightweight approach to introduce an inductive bias into pre-trained language models to capture the hierarchical conversational structure in the source documents. Concretely, RAMMER replaces the full attention mechanism of Transformer (Vaswani et al., 2017) with sparse attention that follows a particular relationship in the conversational structure (e.g., the parent-child relation). To further improve the quality of generated meta-reviews by utilising the metadata information, RAMMER is trained with a multi-task objective to additionally predict source document types, review ratings/confidences and the paper acceptance outcome.

We conduct experiments to compare the performance of RAMMER with a number of baseline models over automatic evaluation metrics including the proposed evaluation metric based on predicting the paper acceptance outcome and human evaluation. We found that RAMMER performs strongly, demonstrating the benefits of incorporating of the conversational structure and the metadata. Further analyses on instances with conflicting source documents, however, reveal that it still struggle to recognise and resolve these conflicts, suggesting that meta-review generation is a challenging task and promising direction for future work.

## 2 Related Work

### 2.1 MDS Datasets

There are a few popular MDS datasets for abstractive summarization in these years, such as WCEP (Ghalandari et al., 2020), Multi-News (Fabbri et al., 2019), Multi-XScience (Lu et al., 2020), and WikiSum (Liu et al., 2018) from news, scientific and Wikipedia domains. Multi-XScience is constructed using the related work section of scientific papers, and takes a paragraph of related work as a summary for the abstracts of its cited papers. Although the summaries are highly abstractive, they are not always reflective of the cited papers — this is attested by the authors' finding that less than half of the statements in the summary are grounded by their source documents. WikiSum and WCEP have a similar problem as they augment source documents with retrieved documents and as such they may only be loosely related to the summary. Notably, none of the source documents in these datasets provides any explicit structure of inter-document relationships or conflicting information, although different inter-document relationships may exist among source documents in these datasets (Ma et al., 2020). This leads to under-explored research on inter-document relationship comprehension of abstractive summarization models. In the peer-review domain, Shen et al. (2022); Wu et al. (2022) developed datasets for meta-review generation. However, they only consider official reviews, or their datasets do not feature the rich hierarchical conversational structure that PEERSUM has.

### 2.2 Structural Inductive Bias for Summarization

Transformer-based pre-trained language models (PLMs) (Lewis et al., 2020; Zhang et al., 2020a; Guo et al., 2022; Phang et al., 2022) are the predominant approach in abstractive text summarization. However, it is challenging to incorporate structural information into the input as Transformer is designed to process flat text sequences. As such, most studies for MDS treat the input documents as a long flat string (via concatenation) without any explicit inter-document relationships (Xiao et al., 2022; Guo et al., 2022; Phang et al., 2022). To take into account the structural information, most work uses graph neural networks (Li et al., 2020; Jin et al., 2020; Cui and Hu, 2021; Li et al., 2023) but it is difficult to construct effective graphs to

| Features | ICLR | NeurIPS |
|---|---|---|
| #samples | 9,835 | 5,158 |
| #official-review-thread/cluster | 3.51 | 3.67 |
| #author-comment-thread/cluster | 0.59 | 0.01 |
| #public-review-thread/cluster | 0.22 | 0.00 |
| #paper-abstract-thread/cluster | 1.0 | 1.0 |

Table 1: PEERSUM statistics.

represent multiple documents and they introduce additional parameters to the pre-trained language models. Attention manipulation is one approach to introduce structural inductive bias without increasing the model size substantially. Studies that take this direction, however, by and large focus on incorporating syntax structure of sentences or internal structure of single documents (Bai et al., 2021; Cao and Wang, 2022) rather than higher level inter-document discourse structure. RAMMER is inspired by these works, and the novelty is that it uses attention manipulation to capture broader inter-document relationships.

## 3 The PEERSUM Dataset

### 3.1 Dataset Construction

PEERSUM is constructed using peer-review data scraped from OpenReview[2] for two international conferences in computer science: ICLR and NeurIPS. As meta-reviewers are supposed to follow the meta-reviewer guidelines[3] with comprehending and carefully summarizing information shown in the peer-reviewing web page (the example shown in Appendix A), and we observe from example meta-reviews as shown in Table 3 that meta-reviewers are complying with the guidelines, we collate the paper abstract, official/public reviews and multi-turn discussions as the source documents, and use the meta-review as the summary. We note that there may be private discussion among the reviewers and meta-reviewer which may influence the meta-review. However, our understanding is that reviewers are advised to amend their reviews if such a discussion changes their initial opinion. For this reason, we believe the meta-review is reflective of the (observable) reviews, discussions and the paper abstract, and this is empirically validated in Section 3.3.

A meta-review (summary) and its corresponding *source documents* (i.e., reviews, discussions and the paper abstract) form a sample in PEERSUM.[4] The source documents has an explicit tree-like conversational structure,[5] as illustrated in Figure 1 (a real example is presented in Appendix A). In total, PEERSUM contains 14,993 samples (train/validation/test: 11,995/1,499/1,499) for ICLR 2018–2022 and NeurIPS 2021–2022; see Table 1 for some statistics. To summarise, PEERSUM has seven types of source documents (shown in different colors in Figure 1): (1) official reviews (reviews by assigned reviewers); (2) public reviews (comments by public users); (3) author comments (an overall response by paper authors); (4) official responses; (5) public responses; (6) author responses within a thread; and (7) the paper abstract. It also features some metadata for each sample: (1) paper acceptance outcome (accept or reject); and (2) a rating (1–10) and confidence (1–5) for each official review.

To compare PEERSUM with other MDS datasets, we present some statistics on sample size and document length for PEERSUM and several other MDS datasets in Table 2.

We next present some analyses to understand the degree of conflicts in the source documents, and abstractiveness and faithfulness in the summaries.

### 3.2 Conflicts in Source Documents

One interesting aspect of PEERSUM is that source documents are not only featuring explicit hierarchical conversational relationships but also presenting conflicting information or viewpoint occasionally such as conflicting sentiments shown in Table 4. We extract *conflicts* among source documents based on review ratings in different official reviews. Denoting CF for samples with conflicts where at least one pair of official reviews that have a rating difference $\geq 4$ (otherwise Non-CF), we found that 13.6% of the dataset are CF samples. The meta-reviews for these instances will need to handle these conflicts. In our experiments (Section 5) we present some results to show whether summarization systems are able to recognize and resolve conflicts in these difficult cases.

---

[2]https://openreview.net/
[3]https://iclr.cc/Conferences/2022/ACGuide, https://nips.cc/Conferences/2022/AC-Guidelines

[4]Henceforth we use the terms *summary* and *meta-review* interchangeably in the context of discussion of PEERSUM.
[5]The average tree height/width = 3.63/5.31.

| Metric | PEERSUM | WikiSum | Multi-News | WCEP | Multi-XScience |
|---|---|---|---|---|---|
| Domain | Peer-review | Wikipedia | News | News | Scientific |
| #Samples | 15,983 | 1,655,709 | 56,216 | 10,200 | 40,528 |
| #Documents/Sample | 10.48 | 40 | 2.79 | 63.38 | 4.45 |
| #Sentences/Document | 19.66 | 2.85 | 30.40 | 18.24 | 7.10 |
| #Tokens/Document | 397.32 | 54.54 | 690.97 | 439.24 | 172.90 |
| #Sentence/Summary | 6.51 | 5.17 | 10.12 | 1.44 | 5.06 |
| #Tokens/Summary | 142.74 | 121.20 | 241.61 | 30.53 | 116.41 |

Table 2: Statistics of PEERSUM and other MDS datasets.

| | |
|---|---|
| M1 | "This meta-review is written after considering the reviews, the authors' responses, the discussion, and the paper itself." |
| M2 | "... the authors made substantial improvements during the discussion phase ..." |
| M3 | "... but the bar for introducing yet another variant of memory-augmented neural nets has been significantly raised, which is a sentiment shared by the reviewers. the author's response had not swayed the reviewers' opinion, and i am sticking to the reviewers' decisions. ..." |

Table 3: Three example meta-reviews (M1, M2, and M3) of meta-review sentences to show that the meta-reviewer is trying to comprehend and carefully summarize information from the paper, the individual reviews, and multi-turn discussions between paper authors and reviewers.

| | |
|---|---|
| P1 | S1: The approach proposed in the paper seems to be a small incremental change on top of the previous GNN pre-train work. *The novelty aspect is low*.
S2: The main contribution is *the novel pre-training strategy* introduced. The work has *potential high impact* in the research area... |
| P2 | S1: Introduction section is *not well-written*.
S2: This paper is *well written* and looks correct. |

Table 4: Two example pairs (P1 and P2) of contradictory sentiments between official reviewers for two scientific papers, and italic texts are conflicts between the two sentences (S1 and S2).

| Dataset | Unigram | Bigram | Trigram |
|---|---|---|---|
| PEERSUM | 28.28 | 82.31 | 92.95 |
| WikiSum | 22.75 | 63.55 | 79.34 |
| Multi-News | 23.49 | 66.10 | 82.01 |
| WCEP | 5.25 | 37.62 | 65.27 |
| Multi-XScience | **44.09** | **86.54** | **96.40** |

Table 5: Percentage of novel n-grams in the summaries of different datasets.

## 3.3 Abstractiveness and Faithfulness of Summaries

Abstractiveness — the degree that a summary contains novel word choices and paraphrases — is an important quality for MDS datasets. Following Fabbri et al. (2019) and Ghalandari et al. (2020), we preprocess source documents and summaries with lemmatisation and stop-word removal, and calculate the percentage of unigrams, bigrams, and trigrams in the summaries that are not found in the source documents and present the results in Table 5. We see that PEERSUM summaries are highly abstractive, particularly for bigrams and trigrams. Although Multi-XScience is the most abstractive, as discussed in Section 2.1 the summaries are not always reflective of the content of source documents.

To understand whether the summaries in PEER-SUM are faithful to the source documents, i.e., whether the statements/assertions in the meta-review are grounded in the source documents, we perform manual analysis to validate this. We recruit 10 volunteers to annotate 60 samples (25 `Non-CF`

and 35 `CF`) to highlight text spans in the summary that can be semantically anchored to the source documents (full instructions for the task is given in Appendix C).[6] Based on the results in Table 6, we can see that for samples with non-conflicting reviews (first row), almost 80% of the words in the meta-reviews are grounded in the source documents. Although this percentage drops to 72% when we are looking at the more difficult cases with conflicting reviews (second row), our analysis reveals that the meta-reviews are by and large faithful, indicating that they function as a good summary of the reviews, discussions and the paper abstract.

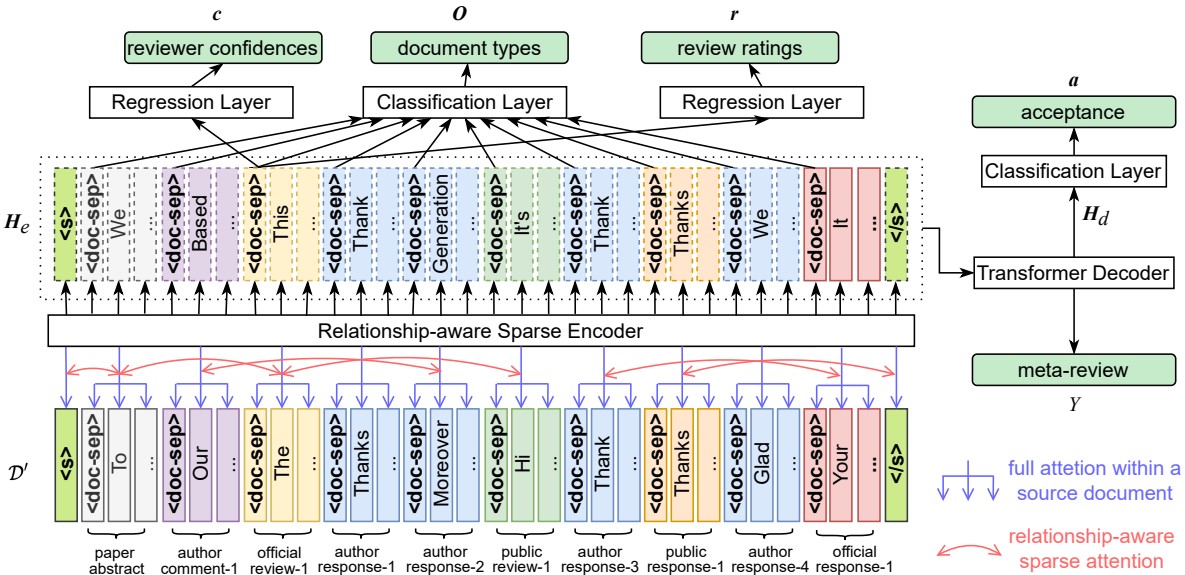

Figure 2: There are six main components in the RAMMER architecture: (1) a relationship-aware sparse encoder to encode source documents; (2) a vanilla Transformer decoder to generate meta-reviews; (3) two different regression layers to predict reviewer confidences and review ratings; (4) two classification layers to predict the type of each source document and the paper acceptance outcome. There are two different types of attention mechanisms: 'full attention within a document' denotes that there are attention calculation between tokens within a document and 'relationship-aware sparse attention' denotes that there are attention calculation between tokens in documents only when there is a connection between the two documents in the corresponding tree-like hierarchical structure.

| Data | #Samples | Mean Variance | Anchored Words (%) |
|---|---|---|---|
| Non-CF | 25 | 0.717 | 79.67% |
| CF | 35 | 6.668 | 72.74% |

Table 6: Percentage of words in the meta-review grounded in the source documents in CF and Non-CF samples. "Mean Variance" denotes the average of rating variance of official reviews.

## 4 The RAMMER Model

We now describe RAMMER, a meta-review generation model that captures the conversational structure in the source documents (Section 4.1) and uses a multi-task objective to leverage metadata information (Section 4.2). RAMMER is built on an encoder-decoder PLM to automatically generate a summary/meta-review $Y$ from a cluster of source documents $\mathcal{D}$; its overall architecture is presented in Figure 2. The input to RAMMER is the concatenation of all source documents ($\mathcal{D}$) and we insert a delimiter <doc-sep> to denote the start of each document.[7]

---

[6]All volunteers are PhD students who major in computer science and are familiar with peer-reviewing.

[7]For PLMs that do not have <doc-sep> in their tokenizers we use  instead.

### 4.1 Relationship-Aware Sparse Attention

To explicitly incorporate hierarchical relationships among source documents into the pretrained Transformer model, we propose an encoder with relationship-aware sparse attention (RSAttn), which improves the summarization performance with the introduction of structural inductive bias. The main idea is to use sparse attention by considering hierarchical conversational relationships among source documents.

Based on the tree-like hierarchical conversational structure and the nature of meta-review generation, we extract seven types of relationships which are represented as matrices (an element is 1 if one document is connected to another, else 0):

- $R_1$, *ancestor-1* which captures the parent asymmetric relationship and the attention from the parent document towards to the current one;
- $R_2$, *ancestor-all* which captures the ancestor asymmetric relationship as the ancestor documents would provide context for the current one;
- $R_3$, *descendant-1* which captures child asymmetric relationship and the attention from the child document towards to the current one;
- $R_4$, *descendant-all* which captures descen-

dant asymmetric relationship as sometimes concerns would be addressed after the discussion in descendant documents;

- $R_5$, *siblings* which captures the sibling symmetric relationship as usually reviewers or the paper authors use sibling documents to provide more complementary information;
- $R_6$, *document-self* which captures the full self-attention among each individual document as token representations are learned based on a rich context within the document;
- $R_7$, *same-thread* which captures the symmetric relationship among documents which are in the same thread (source documents in each grey dashed rectangle in Figure 1) as usually documents in the same thread are talking about the same content.

Next, we use a weighted combination of these relationship matrices to mask out connections to those source documents not included in any relationships for each source document and scale the attention weights. The output of each head in RSAttn in the $l$-th layer is calculated as:

$$H_l = \text{softmax}(\frac{QK^T \odot \sum_j \beta_j \cdot R_j^{\dagger}}{\sqrt{d_k}})V, \quad (1)$$

where $Q$, $K$, and $V$ are representations after the non-linear transformation of $H_{l-1}$, the output of the previous layer, or $X$, the output of the embedding layer from the input $\mathcal{D}'$ with delimiter tokens; $\beta$ is a very small-scale trainable balancing weight vector for different relationships initialized with a uniform distribution, and different heads have different $\beta$, as different heads in each layer may focus on different relationships; $R_j^{\dagger}$ is automatically extended from $R_j$ (if an element of $R_{j,p,q}$ is 1, elements of $R_j^{\dagger}$ from tokens of the $p$-th document to tokens of the $q$-th one are 1, else 0.).

To reduce memory consumption, we implement masking with matrix block multiplication instead of whole attention masking matrices, which means that we only calculate attention weights between every two documents that have at least one relation. This makes the model work for long source documents without substantially increasing computation complexity.

## 4.2 Multi-Task Learning

To utilise metadata information in PEERSUM — review rating, review confidence, paper acceptance outcome and source document type (Section 3.1) —

we train RAMMER on four auxiliary tasks. We use the output embeddings from the encoder to predict review ratings/confidences and source document types, and the output embeddings from the decoder to predict the paper acceptance outcome. Formally, the overall training objective is:

$$\mathcal{L} = \alpha_g \mathcal{L}_g + \alpha_c \mathcal{L}_c + \alpha_r \mathcal{L}_r + \alpha_o \mathcal{L}_o + \alpha_a \mathcal{L}_a \quad (2)$$

where $\alpha$ is used to balance different objectives, $\mathcal{L}_g$ the standard cross-entropy loss for text generation based on the reference meta-review, $\{\mathcal{L}_c, \mathcal{L}_r\}$ the mean squared error for predicting the review confidence and review rating respectively, and $\{\mathcal{L}_o, \mathcal{L}_a\}$ the cross-entropy loss for predicting the paper acceptance outcome and the source document type respectively. Next, we describe more details about auxiliary tasks for the encoder and the decoder.

### 4.2.1 Encoder Auxiliary Tasks

We use $\mathcal{I}^d$ to denote the set of indices containing the special delimiters in the input. Auxiliary objectives of multi-task learning for the encoder are then based on the embeddings of these delimiters. Denoting the output embeddings produced by RAMMER's encoder as $H_e$, we use two regression layers to predict the review confidences $\hat{c}$ and review ratings $\hat{r}$ respectively:

$$\hat{c}_i = \text{sigmoid}(\text{MLP}(H_e^{\mathcal{I}_i^d})), \quad (3)$$

$$\hat{r}_i = \text{sigmoid}(\text{MLP}(H_e^{\mathcal{I}_i^d})) \quad (4)$$

where $\mathcal{I}_i^d$ denotes the index of the delimiter token of the $i$-th official review, and $H_e^{\mathcal{I}_i^d}$ denotes the corresponding embedding. $\mathcal{L}_c$ and $\mathcal{L}_r$ are then computed as:

$$\mathcal{L}_c = \text{mse}(\hat{c}, c), \quad \mathcal{L}_r = \text{mse}(\hat{r}, r), \quad (5)$$

where mse denotes mean squared error, and $c$ and $r$ denote the *normalised* ($[0-1]$) vector of ground truth review confidences and the review ratings, respectively.

To predict the types of source documents we apply a classification layer on the contextual embeddings of its delimiter tokens. The predicted classification distribution $\hat{O}_j$ of the $j$-th source document is computed as follows:

$$\hat{O}_j = \text{softmax}(\text{MLP}(H_e^{\mathcal{I}_j^d})), \quad (6)$$

where $H_e^{\mathcal{I}_j^d}$ denotes the embedding of the $j$-th source document and $\mathcal{I}_j^d$ is the corresponding index

| Initialization | R-L | BERTS | ACC |
|---|---|---|---|
| BART | 27.51 | 15.57 | 0.738 |
| PRIMERA | 29.30 | 13.24 | 0.745 |
| LED | 30.31 | 17.35 | 0.759 |

Table 7: RAMMER performance when initialized with different pre-trained language models.

in $\mathcal{I}^d$. The total loss for predicting all the document types, $\mathcal{L}_o$, is:

$$\mathcal{L}_o = \frac{1}{|\mathcal{I}^d|} \sum_{j=1}^{|\mathcal{I}^d|} \text{cross-entropy}(\boldsymbol{O}_j, \hat{\boldsymbol{O}}_j), \quad (7)$$

where $\boldsymbol{O}_j$ is the one-hot embedding of the ground truth document type of the $j$-th source document.

### 4.2.2 Decoder Auxiliary Tasks

There is only one auxiliary objective for the decoder, to predict the paper acceptance outcome (accept vs. reject):

$$\hat{\boldsymbol{a}} = \text{MLP}(\text{mean}(\boldsymbol{H}_d)), \quad (8)$$
$$\mathcal{L}_a = \text{cross-entropy}(\hat{\boldsymbol{a}}, \boldsymbol{a}), \quad (9)$$

where $\boldsymbol{H}_d$ is the output embeddings from the last layer of the decoder and $\boldsymbol{a}$ is the one-hot embedding of the ground truth paper acceptance.

## 5 Experiments

### 5.1 Experimental Setup

We compare RAMMER with a suite of strong abstractive text summarization models.[8] We have three groups of models that target different types of summarization:[9] (1) short single-document: **BART** (Lewis et al., 2020) and **PEGASUS** (Zhang et al., 2020a); (2) long single-document: **LED** (Beltagy et al., 2020) and **PegasusX** (Phang et al., 2022); and (3) multi-document: **PRIMERA** (Xiao et al., 2022). We use the large variant for these models (which have a similar number of parameters). We fine-tune these models on PEERSUM using the default recommended hyper-parameter settings. All models have the same maximum output tokens (512), but they feature different budgets of the maximum input length. Given a sequence length budget, for each sample we divide the budget by the total number of source documents to get the maximum length permitted for

[8] All experiments are run on 4 NVIDIA 80G A100 GPUs.
[9] We fine-tune these pre-trained models on PEERSUM with the Huggingface library (https://huggingface.co/).

each document and truncate each document based on that length. During training of RAMMER, we use a batch size of 128 with gradient accumulation and label smoothing of 0.1 (Müller et al., 2019). We tune RAMMER's $\alpha$ (Section 4.2) using the validation partition and the optimal configuration is: $\alpha_g = 2, \alpha_c = 2, \alpha_r = 1, \alpha_o = 1, \alpha_a = 2$, indicating that all metadata benefit the final performance and the reviewer confidence and paper acceptance outcome are the more important features. We present more details on training and hyper-parameter configuration in Appendix B.

### 5.2 Automatic Evaluation on Generated Meta-Reviews

We evaluate the quality of generated meta-reviews with metrics including ROUGE (Lin and Hovy, 2003),[10] BERTScore (Zhang et al., 2020b)[11] and UniEval (Zhong et al., 2022)[12]. ROUGE and BERTScore measure the lexical overlap between the generated and ground truth summary, but the former uses surface word forms and latter contextual embeddings. UniEval achieves fine-grained evaluation for abstractive summarization and it is based on framing evaluation of text generation as a boolean question answering task. As faithfulness and informativeness are more important to summarization, we only report the evaluation results of "consistency" and "relevance" from UniEval, respectively.

In addition to these metrics, we introduce another evaluation metric (ACC) based on the metadata of PEERSUM. It is an alternative reference-free metric that measures how well generated meta-reviews are consistent with the ground truth meta-reviews. To this end, we first fine-tune a BERT-based classifier using *ground truth* meta-reviews and paper acceptance outcomes, and then use this classifier to predict the paper acceptance outcome using *generated* meta-reviews. The idea is that if the generated meta-review is consistent with the ground truth meta-review, the predicted paper acceptance outcome should match the ground truth paper acceptance outcome.

As RAMMER can use any encoder-decoder pre-trained models as the backbone, we first present *val-*

[10] For ROUGE-L, we use the summary-level version 'RougeLsum' from https://pypi.org/project/rouge-score/.
[11] Following Koto et al. (2020), we use F1 metrics of ROUGE and BERTScore.
[12] https://github.com/maszhongming/UniEval

| Model(#Params) | Test Data | R-L↑ | BERTS↑ | UniEval-Con↑ | UniEval-Rel↑ | ACC↑ |
|---|---|---|---|---|---|---|
| BART (406M) | Non-CF | 27.50 | 16.61 | 72.97 | 79.87 | 0.728 |
| PEGASUS (568M) | Non-CF | 27.24 | 14.75 | 74.52 | 80.78 | 0.725 |
| PRIMERA (447M) | Non-CF | 28.70 | 12.67 | 68.56 | 82.33 | 0.725 |
| LED (459M) | Non-CF | 29.52 | 16.59 | 70.98 | 82.97 | 0.748 |
| PegasusX (568M) | Non-CF | 29.65 | 17.36 | 73.44 | 82.24 | 0.745 |
| RAMMER (459M) | Non-CF | **30.39*** | **17.42*** | **75.07*** | **83.84*** | **0.768** |
| BART (406M) | CF | 26.84 | 14.89 | 71.85 | 78.74 | 0.683 |
| PEGASUS (568M) | CF | 26.77 | 13.66 | 73.12 | 79.49 | 0.649 |
| PRIMERA (447M) | CF | 29.13 | 12.33 | 66.85 | 81.70 | 0.639 |
| LED (459M) | CF | 29.19 | 15.32 | 70.04 | 82.82 | 0.698 |
| PegasusX (568M) | CF | **29.30** | 15.69 | 71.33 | 81.30 | 0.707 |
| RAMMER (459M) | CF | 29.19 | **15.88*** | **73.21*** | **83.15*** | **0.724** |
| RAMMER (459M) | CF ∪ Non-CF | 30.23 | 17.21 | 74.82 | 83.75 | 0.762 |
| w/o RSAttn (406M) | CF ∪ Non-CF | 29.67 | 16.88 | 71.36 | 83.01 | 0.758 |
| w/o multi-task (406M) | CF ∪ Non-CF | 30.27 | 17.01 | 72.99 | 83.57 | 0.749 |

Table 8: Performance of summarization models over PEERSUM in terms of ROUGE-L F1 (R-L), BERTScore F1 (BERTS), UniEval consistency (UniEval-Con) and relevance (UniEval-Rel) and paper outcome (ACC). Higher value means better performance for all metrics. Results of ROUGE-1 and ROUGE-2 which are not present are consistent with that of ROUGE-L. *: significantly better than others in the same group (p-value < 0.05).

*idation* results for RAMMER where it is initialised with BART, PRIMERA and LED with different maximum input lengths (1,024, 4,096 and 4,096, respectively) in Table 7. We see consistently that the LED variant performs better than the other two, and this helps us choose the LED variant as the backbone of RAMMER. This also indicates that our idea of RSAttn and multi-task learning with metadata can also work on other pre-trained language models.

We next compare RAMMER with the baseline text summarization models on the *test* set in Table 8[13]. Here we also break the test partition into CF and Non-CF samples. Broadly speaking, summarisation performance across all metrics for CF is lower than that of Non-CF, confirming our suspicion that the CF instances are more difficult to summarise. The disparity is especially significant for ACC and BERTScore, suggesting that these two are perhaps the better metrics for evaluate the quality of generated meta-reviews. Comparing RAMMER with the baselines, it is encouraging to see that it is consistently better (exception: R-L results of most models on CF samples are more or less the same). This demonstrates the benefits of incorporating the conversational structure and metadata in the source documents into pre-trained language models. To better understand the impact of RAMMER's sparse

attention (RSAttn; Section 4.1) and multi-task objective (Section 4.2), we also present two RAMMER ablation variants (the last three rows in Table 8). It is an open question which method has more impact, as even though most metrics (R-L, BERTScore and UniEval) seem to indicate RSAttn is the winner, ACC — which we believe is the most reliable metric — appear to suggest otherwise. That said, we can see they complement with each other and as such incorporating both produces the best performance.

## 5.3 Human Evaluation on Conflict Recognition and Resolution

To dive deeper into understanding how well these summarization models recognize and resolve conflicting information in source documents, we conduct a human evaluation.

We randomly select 40 CF samples and recruit two volunteers[14]. We ask them to first assess whether each *ground truth* meta-review **recognises** conflicts, i.e., whether the meta-review discusses or mentions conflicting information/viewpoints that are in the official reviews. For each sample, the volunteers are presented with all the source documents and are asked to make a binary judgement about conflict recognition. We found that 23 out of 40 ground truth meta-reviews have successfully

---

[13]Some random examples and corresponding model generations are present in Appendix D.

[14]Both volunteers major in computer science and are familiar with peer-reviewing.

| Model | Recognition | Resolution |
|---|---|---|
| PRIMERA | 3/23 | 2/23 |
| LED | 4/23 | 4/23 |
| PegasusX | 5/23 | 5/23 |
| RAMMER | 8/23 | 3/23 |

Table 9: Performances of summarization models on conflict recognition and resolution for CF samples.

done this, and we next focus on assessing generated meta-reviews for these remaining 23 samples.

For these 23 samples, we ask the volunteers to assess conflict recognition for *generated* meta-reviews. Additionally, we also ask them to judge (binary judgement) whether the generated meta-review **resolves** the conflicts in a similar manner consistent with the ground truth meta-review. Conflict recognition and revolution results for RAMMER and three other baselines are presented in Table 9. In terms of recognition, relatively speaking RAMMER does better than the baselines which is encouraging, but ultimately it still fails to recognise conflicts in majority of the samples. When it comes to conflict resolution, all the models perform very poorly, indicating the challenging nature of resolving conflicts in source documents of PEERSUM.

## 6 Conclusion

We introduce PEERSUM, an MDS dataset for meta-review generation. PEERSUM is unique in that the summaries (meta-reviews) are grounded in the source documents despite being highly abstractive, it has a rich set of metadata and explicit inter-document structure, and it features explicit conflicting information in source documents that the summaries have to handle. In terms of modelling, we propose RAMMER, an approach that extends Transformer-based pre-trained encoder-decoder models to capture inter-document relationships (through the sparse attention) and metadata information (through the multi-task objective). Although RAMMER is designed for meta-review generation here, our approach of manipulating attention to incorporate the input structure can be easily adapted to other tasks where the input has inter-document relationships. Compared with baselines over a suite of automatic metrics and human evaluation, we found that RAMMER performs favourably, outperforming most strong baselines consistently. That said, when we assess how well RAMMER does for situations where there are conflicting informa-

tion/viewpoints in the source documents, the outlook is less encouraging. We found that RAMMER fail to recognise and resolve these conflicts in its meta-reviews in the vast majority of cases, suggesting this is a challenging problem and promising avenue for further research.

## Limitations

Our work frames meta-review generation as an MDS problem, but one could argue that writing a meta-review requires not just summarising key points from the reviews, discussions and the paper abstract but also wisdom from the meta-reviewer to judge opinions. We do not disagree, and to understand the extent to which the meta-review can be "generated" based on the source documents we conduct human assessment (Section 3.3) to validate this. While the results are encouraging (as we found that most of the content in the meta-reviews are grounded in the source documents) the approach we took is a simple one, and the assessment task can be further improved by decomposing it into subtasks that are more objective (e.g., by explicitly asking annotators to link statements in the meta-reviews to sentences in the source documents).

In the age of ChatGPT and large language models, there is also a lack of inclusion of larger models for comparison. We do not believe it makes sense to include closed-source models such as ChatGPT for comparison (as it is very possible that they have been trained on OpenReview data), but it could be interesting to experiment with large open-source models such as OPT (Zhang et al., 2022), LLaMA (Touvron et al., 2023) or Falcon (Almazrouei et al., 2023). We contend, however, the results we present constitute preliminary results, and that it could be promising direction to explore how RAMMER's RSAttn can be adapted for large autoregressive models.

Lastly, we only consider explicit conversational structure in this paper. As our results show, incorporating such structure only helps to recognise conflicts to some degree but not for resolving them. It would be fascinating to test if incorporating *implicit* structure, such as argument and discourse links, would help. This is not explored in this paper, but it would not be difficult to adapt our methods to incorporate these structures.

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

# A   Appendix: Different types of source documents in PEERSUM

We present a real PEERSUM example with annotation in Figure 3. Please note that we randomly select this example from PEERSUM and we have removed the author names of the paper and reviewer names, and the content in this real example is not the same as in the synthesized example in Figure 1, while both of the two feature hierarchical inter-document relationships. As shown in Figure 3, automatic meta-review generation is aiming to generate the meta-review automatically based on the paper abstract, official and public reviews and the multi-turn discussions.

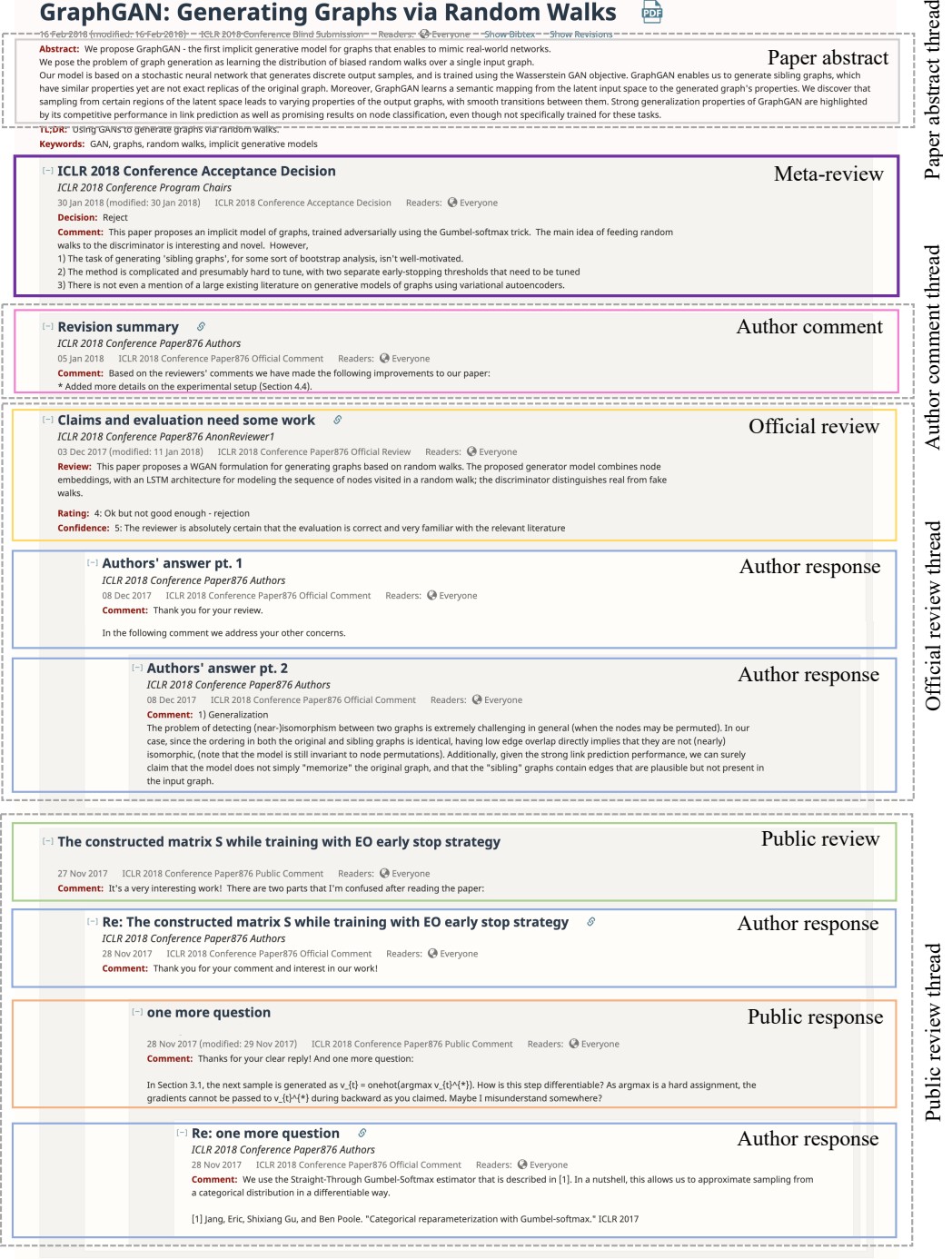

Figure 3: A set of source documents and the corresponding meta-review for a scientific paper in PEERSUM.

## B   Appendix: Hyper-parameters for fine-tuning pre-trained text summarization models

We present all hyper-parameters in Table 10 for all models.

| Model | Max-len(in/out) | optimizer | lr | warm up | scheduler | batch size | beam size | length penalty |
|---|---|---|---|---|---|---|---|---|
| BART | 1,024/512 | Adafactor | 3e-5 | 0k | constant schedule | 64 | 5 | 1.0 |
| PEGASUS | 1,024/512 | Adafactor | 5e-5 | 0k | square root decay | 256 | 8 | 0.8 |
| PRIMERA | 4,096/512 | Adam | 3e-5 | 0.5k | linear decay | 16 | 5 | 1.0 |
| LED | 4,096/512 | Adam | 3e-5 | 0.2k | linear decay | 32 | 5 | 0.8 |
| PegasusX | 4,096/512 | Adafactor | 8e-4 | 0k | constant schedule | 64 | 1 | 1.0 |
| RAMMER | 4,096/512 | Adafactor | 5e-5 | 0.2k | linear decay | 128 | 5 | 1.0 |

Table 10: Hyper-parameters for all models in experiments.

## C   Appendix: Instructions for annotation of PEERSUM quality

Welcome to the annotation project for PeerSum. Please have a careful read of the project introduction and task instructions and finish the tasks in the separate document.

**Introduction of the project:**

To enhance the capabilities of multi-document summarization systems we present PeerSum, a novel dataset for automatically generating meta-reviews of scientific papers based on reviews, multi-turn discussions and the paper abstract in the peer-reviewing process in https://openreview.net/. In the reviewing process, all assigned reviewers and public users can give comments to each paper, and then the author of the paper might respond to those comments. There may be a couple of rounds of discussions or rebuttals during the reviewing process. In the end, the meta-reviewer will write a summary of these comments and discussions, to support their final decision on the paper acceptance. Usually, meta-reviewers are supposed to write the meta-review based on summarizing all reviews, discussions, and the paper abstract, but they may sometimes draw on some external knowledge which is not present in the source documents, such as their own knowledge in the field, reading of the full paper beyond the paper abstract.

The objective of the annotation task is to assess whether the statements/assertions in meta-reviews are exclusively drawn from the reviews, discussion and the paper abstract which are the source documents in PeerSum. Annotators are expected to help highlight the statements/assertions in meta-reviews that can be drawn from the source documents. Highlighted texts will be heavily dependent on source documents and mainly talk about information that is present in the source documents, while they will not be heavily dependent on meta-reviewer's judgements, the meta-reviewer's own knowledge in the field, reading of the full paper beyond the paper abstract, or any other external knowledge relative to the source documents. Please note that if assertions have very light judgement from meta-reviewers but the content are mostly drawn from source documents, we will prefer to highlight these assertions, as these assertions usually cover much about critical information in the source documents.

**Instructions for the task:**

Each of you will get 6 samples in total. For each sample:

- Please carefully read the source documents including the paper abstract, reviews by different reviewers, and discussions between reviewers and the author (all responses) in the linked OpenReview page.
- Please read the meta-review which is the same as the section of Paper Decision in the corresponding OpenReview link, and highlight all assertions or statements (which may be a clause, a sentence, or a paragraph) which draws knowledge solely from source documents with the colour of blue.

**Annotation examples:**

Please also carefully read the following two examples of annotation tasks. We also prepare explanations for unhighlighted or highlighted texts following each example, but you do not need to write explanations when annotating.

**Example one**

Source Documents:

Link to OpenReview: `https://github.com/oaimli/PeerSum/blob/main/examples/Hygy01StvH.pdf`

Meta-review:

The reviewers have pointed out several major deficiencies of the paper, which the authors decided not to address.

**Example two**

Source Documents:

Link to OpenReview: `https://github.com/oaimli/PeerSum/blob/main/examples/H1DkN7ZCZ.pdf`

Meta-review:

Authors present a method for representing DNA sequence reads as one-hot encoded vectors, with genomic context (expected original human sequence), read sequence, and CIGAR string (match operation encoding) concatenated as a single input into the framework. Method is developed on 5 lung cancer patients and 4 melanoma patients. Pros: - The approach to feature encoding and network construction for task seems new. – The target task is important and may carry significant benefit for healthcare and disease screening. Cons: - The number of patients involved in the study is exceedingly small. Though many samples were drawn from these patients, pattern discovery may not be generalizable across larger populations. Though the difficulty in acquiring this type of data is noted. – (Significant) Reviewer asked for use of public benchmark dataset, for which authors have declined to use since the benchmark was not targeted toward task of ultra-low VAFs. However, perhaps authors could have sourced genetic data from these recommended public repositories to create synthetic scenarios, which would enable the broader research community to directly compare against the methods presented here. The use of only private datasets is concerning regarding the future impact of this work. – (Significant) The concatenation of the rows is slightly confusing. It is unclear why these were concatenated along the column dimension, rather than being input as multiple channels. This question doesn't seem to be addressed in the paper. Given the pros and cons, the committee recommends this interesting paper for workshop.

Explanations:

- "However, perhaps authors could have sourced genetic data from these recommended public repositories to create synthetic scenarios," is highlighted, because this assertion is logically based on recommended public repositories and synthetic scenarios which are from source documents.
- "which would enable the broader research community to directly compare against the methods presented here. The use of only private datasets is concerning regarding the future impact of this work." is not highlighted, because this is heavily based on meta-reviewer's own experience in the field or suggestion about impact of the paper.
- "This question doesn't seem to be addressed in the paper." is not highlighted, because it is a meta-reviewer's own judgement about the paper.
- In "Given the pros and cons, the committee recommends this interesting paper for workshop." which is not highlighted, there is external knowledge about the workshop information.

**Example three**

Source Documents:

Link to OpenReview: `https://github.com/oaimli/PeerSum/blob/main/examples/ZeE81SFTsl.pdf`

Meta-review:

Dear authors, I apologize to the authors for insufficient discussion in the discussion period. Thanks for carefully responding to reviewers. Nevertheless, I have read the paper as well, and the situation is clear to me (even without further discussion). I will not summarize what the paper is about, but will instead mention some of the key issues. 1) The proposed idea is simple, and in fact, it has been known to me for a number of years. I did not think it was worth publishing. This on its own is not a reason for rejection, but I wanted to mention this anyway to convey the idea that I consider this work very incremental. 2) The

idea is not supported by any convergence theory. Hence, it remains a heuristic, which the authors admit. In such a case, the paper should be judged by its practical performance, novelty and efficacy of ideas, and the strength of the empirical results, rather than on the theory. However, these parts of the paper remain lacking compared to the standard one would expect from an ICLR paper. 3) Several elements of the ideas behind this work existed in the literature already (e.g., adaptive quantization, time-varying quantization, ...). Reviewers have noticed this. 4) The authors compare to fixed / non-adaptive quantization strategies which have already been surpassed in subsequent work. Indeed, QSGD was developed 4 years ago. The quantizers of Horvath et al in the natural compression/natural dithering family have exponentially better variance for any given number of levels. This baseline, which does not use any adaptivity, should be better, I believe, to what the author propose. If not, a comparison is needed. 5) FedAvg is not the theoretical nor practical SOTA method for the problem the authors are solving. Faster and more communication efficient methods exist. For example, method based on error feedback (e.g., the works of Stich, Koloskova and others), MARINA method (Gorbunov et al), SCAFFOLD (Karimireddy et al) and so on. All can be combined with quantization. 6) The reviewer who assigned this paper score 8 was least confident. I did not find any comments in the review of this reviewer that would sufficiently justify the high score. The review was brief and not very informative to me as the AC. All other reviewers were inclined to reject the paper. 7) There are issues in the mathematics – although the mathematics is simple and not the key of the paper. This needs to be thoroughly revised. Some answers were given in author response. 8) Why should expected variance be a good measure? Did you try to break this measure? That is, did you try to construct problems for which this measure would work worse than the worst case variance? Because of the above, and additional reasons mentioned in the reviewers, I have no other option but to reject the paper. Area Chair

Explanations:

- In this meta-review, the meta-reviewer write it based on own reading of the full paper. In this kind of cases, meta-reviewers draw on external knowledge, but some of the assertions are still based on source documents, such as "All other reviewers were inclined to reject the paper".
- "Dear authors, I apologize to the authors for insufficient discussion in the discussion period. Thanks for carefully responding to reviewers." is not highlighted, because this is coordination words and some own judgements from the meta-reviewer.
- "Nevertheless, I have read the paper as well, and the situation is clear to me (even without further discussion)." is not highlighted, because this is based on meta-reviewer's own reading of the full paper.
- "I will not summarize what the paper is about, but will instead mention some of the key issues." is not highlighted, because this is coordination words from the meta-reviewer.
- "1) The proposed idea is simple, and in fact, it has been known to me for a number of years. I did not think it was worth publishing. This on its own is not a reason for rejection, but I wanted to mention this anyway to convey the idea that I consider this work very incremental." is not highlighted, because this is based on the meta-reviewer's own experience.
- "In such a case, the paper should be judged by its practical performance, novelty and efficacy of ideas, and the strength of the empirical results, rather than on the theory. However, these parts of the paper remain lacking compared to the standard one would expect from an ICLR paper." is not highlighted, because this is the meta-reviewer's experience about the standard of ICLR.
- "which have already been surpassed in subsequent work. Indeed, QSGD was developed 4 years ago. The quantizers of Horvath et al in the natural compression/natural dithering family have exponentially better variance for any given number of levels. This baseline, which does not use any adaptivity, should be better, I believe, to what the author propose. If not, a comparison is needed." is not highlighted, because this is based on the meta-reviewer's experience in the field.
- "5) FedAvg is not the theoretical nor practical SOTA method for the problem the authors are solving. Faster and more communication efficient methods exist. For example, method based on error feedback (e.g., the works of Stich, Koloskova and others), MARINA method (Gorbunov et al), SCAFFOLD (Karimireddy et al) and so on. All can be combined with quantization." is not

highlighted, because this is based on the meta-reviewer's experience in the field.

- "I did not find any comments in the review of this reviewer that would sufficiently justify the high score. The review was brief and not very informative to me as the AC." is not highlighted, because this is meta-reviewer's judgement on the review.
- "7) There are issues in the mathematics – although the mathematics is simple and not the key of the paper. This needs to be thoroughly revised.", this is not highlighted because it is based on meta-reviewer's reading of the full paper.
- "8) Why should expected variance be a good measure? Did you try to break this measure? That is, did you try to construct problems for which this measure would work worse than the worst case variance? Because of the above, and", this is not highlighted because it is based on the meta-reviewer's own knowledge in the field.
- "I have no other option but to reject the paper. Area Chair" is not highlighted, as this is the meta-reviewer's judgement on the paper.

## D  Appendix: Generated meta-reviews for PEERSUM by different models

We present five groups of example meta-reviews generated by fully-supervised PRIMERA, LED, Pega-susX, and RAMMER in Table 11 with the input of varying lengths, 1,024, 4,096, and 4,096, respectively, and also ROUGE scores measuring the quality of generated meta-reviews in comparison to the ground truth one. These examples are randomly selected from the test set of PEERSUM. It is clear to see that although RAMMER outperforms other strong baseline models in terms of evaluation metrics in Table 8, the quality of generated meta-reviews still needs to be improved. This further confirms our claim that PEERSUM is a really challenging dataset.

---

Example 1, https://github.com/oaimli/PeerSum/blob/main/examples/e1.pdf

---

| | |
|---|---|
| *Ground truth meta-review* | Understanding neural networks once they have been trained is a big open problem for machine learning. This manuscript designed graph theoretic and information theoretic measures aimed at helping us understand community structure and function in trained networks. In particular, they measure community structure (modularity) and entropy for trained networks and related these to the performance of the networks. The manuscript runs experiments with fully connected networks on problems such as MNIST and CIFAR. Both community structure and entropy measures are shown to correlate (Spearman and Pearson correlation coefficients) with performance metrics in the networks studied. Reviewers tended to agree that the paper was well written and motivated by an interesting and timely question (understanding trained networks). However, on the whole, most of the reviewers believe that the manuscript is too preliminary for publication at ICLR and I agree. A central issue cited by most of the reviewers is that the experiments are performed on small/toy models for small tasks and under particular hyperparameter regimes. It is therefore unclear to what extent the results would generalize to other situations. E.g. would the results hold for larger dataset or for convolutional neural networks? Connected to this complaint, reviewers worry that there is not enough connection to the literature and baseline methods that could be used to predict performance given measures of trained network activity. Even allowing that the observed correlations are true and generalizable, are these measures better than those covered elsewhere in the literature? Additionally problematic, the measures are not theoretically justified either. Thus, we are missing both reasoned arguments for the metrics and robust quantification beyond a limitted experimental setting. One reviewer, Xmnm, is compelled by the work and recommends acceptance. However, they do not present a compelling case for acceptance, and even repeat several of the concerns raised by other reviewers. In sum, the work is on an interesting subject and timely, but needs further work to be ready for publication. |
| *PRIMERA* | This paper proposes two methods to analyze the behavior of neurons in neural nets. The main idea is to study the neuron activation patterns of classification models and explore if the performance can be explained through neurons' activation behavior. The authors propose two approaches: one that models neurons' activation behavior as a graph and examines whether the neurons form meaningful communities, and the other that examines the predictability of neurons' behavior using entropy. The paper is interesting, easy to read, and very well-structured. Furthermore, the subject of the paper is very relevant: the exploration of tools to increase the understanding and explainability of neural models. However, several major concerns are obvious: (1) The contribution of this paper is not properly compared against existing methods, making it difficult to tell their significance. (2) The scalability of this work is suspicious. There 's no guarantee that proposed activation analysis methods will also show the same correlation tendency in other neural networks with more complicated architectures than multi-layer perceptron networks. (3) Experiments on other various hyperparameters/conditions that can affect neuron configurations or learning ability of the neural network(e.g., pruning) could add more validities to their 'comprehensive experimental study'. (4) The suggested explanatory methodologies and approaches were pretty persuasive to explain the network behavior in microscopic context. Nevertheless, the scalability and practicality of the work should be examined carefully. (ROUGE-1: 0.37184, ROUGE-2: 0.04348, ROUGE-L: 0.34657) |

| | |
|---|---|
| *LED* | This paper proposes two new metrics to analyze the behavior of neurons in neural nets. The main idea is to study the neuron activationpatterns of classification models and explore if the performance can be explained through neurons' activation behavior. The authors propose two approaches: one that models neurons' activation behavior as a graph and examines whether the neurons form meaningful communities, and the other that examines the predictability of neurons' behavior using entropy. The reviewers agree that this is an interesting paper and the authors have done a good job in their rebuttal. However, the paper is not ready for publication at ICLR due to the following reasons: 1. There are many existing approaches in explaining deep learning models, such as turning deep models into decision trees in "Distilling a Neural Network Into a Soft Decision Tree". Then complexity measurement of the decision tree can be a baseline. It would be interesting to see how the techniques presented in this paper compare to other techniques already present in the literature. (ROUGE-1: 0.36032, ROUGE-2: 0.07317, ROUGE-L: 0.34008) |
| *PegasusX* | The paper proposes two methods to analyze the behavior of neurons in neural nets. The main idea is to study the neuron activation patterns of classification models and explore if the performance can be explained through neurons' activation behavior. The authors propose two approaches: one that models neurons' Activation behavior as a graph and examines whether the neurons form meaningful communities, and the other that examines the predictability of neurons' behavior using entropy. The paper is interesting, easy to read, and well-structured. However, the paper is below the acceptance threshold. The reviewers raised several concerns, including the lack of a thorough literature survey, the lack and limited experimental validation, and concerns about the scalability of the proposed methods. The AC agrees with the reviewers and recommends rejection. (ROUGE-1: 0.32743, ROUGE-2: 0.03111, ROUGE-L: 0.29646) |
| RAMMER | The paper proposes two methods to analyze the behavior of neurons in neural nets. The authors propose two approaches: one that models neurons' activation behavior as a graph and examines whether the neurons form meaningful communities, and the other that examines the predictability of neurons' behavior using entropy. They found that entropy is negatively correlated withtraining accuracy, modularity is correlated with training accuracy, and that the number of well defined neural communities increase with accuracy. The reviewers agree that the paper is interesting, easy to read, and very well-structured. The main technical contribution of this paper comes from explaining the neural classifiers by combining the graph-theoretic and information- theoretic approaches. The reviewers pointed out that the entropy based performance metric is a novel contribution, which sets our work apart from previous researches. However, the scalability and practicality of the work should be examined carefully. (ROUGE-1: 0.35319, ROUGE-2: 0.06410, ROUGE-L: 0.34043) |

Example 2, https://github.com/oaimli/PeerSum/blob/main/examples/e2.pdf

| | |
|---|---|
| *Ground truth meta-review* | This paper studies the method to achieve the batch size-invariant for policy gradient algorithms (PPO, PPG). The paper achieves this by decoupling the proximal policy from the behavior policy. Empirical results show that the methods are somewhat effective at providing batch size invariance.

After reading the authors' feedback, the reviewer discussed the paper and they did not reach a consensus. On the one hand, the rebuttal made some reviewers change their minds who appreciated the explanations provided by the authors and the new Figure that better highlights the batch size invariance property. On the other hand, some reviewers think that there is still significant work to be done to get this paper ready for publication. In particular, it is necessary to improve the theoretical analysis and the evaluation of the empirical results.

I encourage the authors to follow the reviewers' suggestions while they will update their paper for a new submission. |
| *PRIMERA* | This paper proposes a new method to deal with batch size-invariance for policy optimization algorithms such as PPO and PPG. The key insight is to decouple the proximal policy (used for controlling policy updates) from the behavior policy ( used for off-policy corrections). Our experiments help explain why these algorithms work, and additionally show how they can make more efficient use of stale data.

The reviewers agree that this paper studies an interesting problem of batch size invariance. The writing is clear, and the method is easy to understand. The experimental results are sufficient to demonstrate the effectiveness of the proposed method.

However, the reviewers also agree that the paper could be improved in terms of clarity of presentation and the design of the experiments. In particular, it is important to provide a more rigorous analysis of our ablations, showing that the effects of all of the ablations are statistically significant at the 0.1
(ROUGE-1: 0.48780, ROUGE-2: 0.17439, ROUGE-L: 0.46612) |
| *LED* | This paper proposes a method to achieve batch size-invariance for policy optimization. The reviewers agree that this is an interesting problem and the paper is well-written. However, the reviewers also agree that the paper lacks theoretical justification, and the claim of the batch size invariant property is vague and weak. The authors are encouraged to improve the paper and resubmit to a future venue.
(ROUGE-1: 0.41284, ROUGE-2: 0.21296, ROUGE-L: 0.40367) |
| *PegasusX* | The paper proposes a method to achieve batch size invariance for policy optimization algorithms such as PPO and PPG. The paper achieves this by decoupling the proximal policy from the behavior policy. The experiments demonstrate the effectiveness of the method. The reviewers were unanimous in their opinion that the paper is not ready for publication at ICLR. The main concerns were around the clarity of the presentation and the experimental design. The authors have addressed some of the concerns in their rebuttal, but the paper still needs significant work before it can be accepted.
(ROUGE-1: 0.52846, ROUGE-2: 0.23770, ROUGE-L: 0.50407) |

| | |
|---|---|
| RAMMER | The paper proposes a method to deal with batch size-invariance for policy optimization algorithms such as PPO and PPG. The paper achieves this by decoupling the proximal policy from the behavior policy. The authors also provide two new algorithm variants to achieve the batch size invariance property. |
| | The reviewers agree that the paper is clearly written, is simple to read, and flows well. The experimental results are sufficient to demonstrate the effectiveness of the proposed method. However, the theoretical analysis on how this method could be batch-invARIant in the surrogate loss or final performance is not sufficient. The reviewers also point out the lack of theoretical justification, especially the use of the EWMA. |
| | (ROUGE-1: 0.54682, ROUGE-2: 0.24151, ROUGE-L: 0.50187) |

Example 3, 

| | |
|---|---|
| *Ground truth meta-review* | This paper reveals that popular data poisoning systems, Fawkes and LowKey, fail to effectively protect user privacy in facial recognition. The methods to defend against poisoning attacks are quite simple—you can either adaptively tune the face recognition models or just wait for more advanced facial recognition systems. Given these "disappointed" findings from the technical solution side, this paper further argues that legislation may be the only viable solution to prevent abuses of facial recognition. |
| | Overall, all the reviewers highly appreciate the comprehensive and rigorous evaluations provided in this paper and enjoy reading it. The biggest concern is raised by the Reviewer 6s7m, given this work fails to discuss/compare to previous works on Facial identity anonymizing and the technical contribution is incremental. During the discussion period, all other reviewers reach a consensus that 1) facial identity anonymizing is not relevant; and 2) this work make enough contributions and is worthy to be heard by the general community; the Reviewer 6s7m still hold the opposite opinion, but is okay for accepting this paper anyway. |
| | In the final version, the authors should include all the clarification provided in the discussion period. |

| | |
|---|---|
| *PRIMERA* | This paper studies the effect of data poisoning in face recognition and the relation to the defense techniques. Two solutions of defense are given: oblivious trainer and adaptive trainer. The claim is that, any existing poisoning methods cannot protect the privacy of users in the face images. The paper shows that perturbing facial images does not offer long term security; future systems can still recognize the once perturbed image(s). 

 The paper is well written and easy to follow. The topic discussed in this paper is of significant importance as the rapid development of deep learning techniques nowadays also poses great threats to people's privacy, especially for the face data. Extensive experiments have been conducted to statistically ascertain the authors' claims. 

 However, the technical novelty is very limited. Rather than fancying legislative alternative, a research paper needs to propose technical solution. The original evaluations performed by the Fawkes and LowKey papers were deemed sufficient by reviewers at the time. We believe our paper thus makes an important contribution in demonstrating how such evaluations may have overestimated the protections offered to users. The design of defense strategies that achieve high robustness and accuracy, in Section 3.5. These strategies demonstrate that robustness need not come at a cost in accuracy in all situations, as is often claimed in the adversarial examples literature. 

 Overall, this is a facial privacy analysis with insightful claims, but the presentation and the discussion is very confusing. Thus, we are not able to agree this argument whether is reasonable and solid. The final rating will depend on the authors' feedback. 
 (ROUGE-1: 0.41410, ROUGE-2: 0.07522, ROUGE-L: 0.38326) |
| *LED* | This paper studies the effect of data poisoning in face recognition and the relation to the defense techniques. It shows that the methods fail over time and they do not generalize to future attacks. Two solutions of defense are given: oblivious trainer and adaptive trainer. Both methods successfully defeat two poisoning attack baselines, raising awareness on the inefficiency of the poisoning-based identity protection mechanism. 

 The paper is well written and easy to follow. The topic discussed in this paper is of significant importance as the rapid development of deep learning techniques nowadays also poses great threats to people's privacy, especially for the face data. It is in interesting paper and has value for the community to consider longer term security measures for facial recognition privacy concerns. The experimental results are within expectation and little technical insight is gained. 
 (ROUGE-1: 0.40606, ROUGE-2: 0.06707, ROUGE-L: 0.35152) |
| *PegasusX* | The paper presents a new defense against data poisoning attacks. The paper is well written and the analysis is thorough. The authors also provide a discussion of the defense and the attack dynamics. The main concern is that the technical novelty is limited. The rebuttal did not convince the reviewers. The AC agrees with the reviewers and recommends acceptance. 
 (ROUGE-1: 0.29719, ROUGE-2: 0.06478, ROUGE-L: 0.28112) |

| RAMMER | The paper studies the problem of data poisoning in face recognition and the relation to the defense techniques. The paper shows that perturbing facial images does not offer long term security; future systems can still recognize the once perturbed image(s). The paper also shows that an "oblivious" model trainer can simply wait for future developments in computer vision to nullify the protection of pictures collected in the past.

The paper is well-written and easy to read. The authors have examined several strategies to enable modern face recognition models to defense attacks from widely used data poisoning methods. However, the technical contribution from this paper is limited as little new insight is provided. The experimental results are within expectation and little technical insight is gained. The main conclusion is that people should not rely on technical solutions to protect users privacy and legislation actions are what is actually needed.
(ROUGE-1: 0.42353, ROUGE-2: 0.08284, ROUGE-L: 0.39412) |
|---|---|

Example 4, https://github.com/oaimli/PeerSum/blob/main/examples/e4.pdf

| *Ground truth meta-review* | This paper presents a method which selects feasible data augmentations suitable for contrastive time series representation learning. The topic in this paper is timely and interesting. One of 4 reviewers did not complete the review, not responding to a few reminders. So, one emergency reviewer, who is an expert in meta-learning was added. While there is one review that strongly supports this work, two reviews remained unsupportive after the discussion period ended. I appreciate the authors for making efforts in responding to reviewers' comments. However, after the discussion period, most of reviewers had concerns in this work, pointing out that the technical correctness needs further justification and experiments should be improved. While the idea is interesting, the paper is not ready for the publication at the current stage. I encourage to resubmit the paper after addressing these concerns. |
|---|---|
| *PRIMERA* | This paper proposes an information-aware approach to representation learning for time series. The formulation focuses on how to obtain effective data augmentations and addresses the underlying problem from information-theoretic viewpoints, leading to the two optimization criteria, namely, high fidelity and high variety. The experimental results on several time series datasets for forecasting and classification show improvements over the methods in comparison.

All reviewers agree that this is a very good submission that touches upon a kind of neglected and less fancy area of data augmentation for time-series - authors propose a very neat solution that improves upon the state of the art and demonstrates good performance across datasets and various ablation studies.
(ROUGE-1: 0.31621, ROUGE-2: 0.03187, ROUGE-L: 0.26877) |
| *LED* | This paper proposes a new approach to data augmentation for time series representation learning. The reviewers agree that the paper is well-written and the proposed approach is novel and interesting. However, there are concerns about the novelty of the approach and the significance of the results. The authors are encouraged to revise the paper based on the reviewers' comments and resubmit to a future venue.
(ROUGE-1: 0.41951, ROUGE-2: 0.15764, ROUGE-L: 0.40000) |

| | |
|---|---|
| *PegasusX* | The paper proposes a new data augmentation approach based on information theory, a meta learning approach and an approach to select optimal data augmentation for contrastive learning. The paper is well written and the proposed approach is interesting. However, the experimental results are not convincing. The authors have provided a detailed response to the reviews, but the paper still needs more work before publication.
(ROUGE-1: 0.34483, ROUGE-2: 0.09950, ROUGE-L: 0.32512) |
| RAMMER | This paper proposes a meta-learning approach for data augmentation for time series. The reviewers found the paper to be well-written and well-motivated, and the approach to be novel and interesting. However, the reviewers also found the experimental results to be insufficiently convincing. In particular, some of the analyses and ablations could be more complete, in order to truly investigate whether the conclusions are correct.
(ROUGE-1: 0.33010, ROUGE-2: 0.06863, ROUGE-L: 0.33010) |

Example 5, https://github.com/oaimli/PeerSum/blob/main/examples/e5.pdf

| | |
|---|---|
| *Ground truth meta-review* | This works presents a new and interesting notion of margin for deep neural networks (that incorporates representation at all layers). It then develops generalization bounds based on the introduced margin. The reviewers pointed some concerns, including some notation issues, complexity in case of residual networks, removal of exponential dependence on depth, and dependence on a hard to compute quantity - $\kapp^{adv}$. Some of these concerns were addressed by the authors. At the end, most of the reviewers find the notion of all-layer margin introduced in this paper a very novel and promising idea for characterizing generalization in deep networks. Agreeing with reviewers, I recommend accept. However, I request the authors to accommodate remaining comments /concerns raised by R1 in the final version of your paper. In particular, in your response to R1 you mentioned for one case you saw improvement even with dropout, but that is not mentioned in the revision; Please include related details in the draft. |
| *PRIMERA* | The paper presents a bound on the generalization error of a deep network in terms of margin at each layer of the network. The starting premise is that extending the existing margin generalization bounds to deep networks worsen exponentially with the depth. Recent work which removed that exponential dependency is claimed to require a more involved proof and complicated dependence on input. The paper provides a new bound that is simpler and tighter. A second contribution is to extend their bounds to robust classifier. Since their bounds depend on instance-specific margins, the extension to the robust case is straightforward. Finally, they present a new algorithm motivated by their bounds, that maximized margin on all layers. They show that the resultant network has much lower error than standard training. The paper is well-presented and in spite of being theoretical is very nicely developed so that the main contributions come out clearly to non-specialists too. A few minor comments: The inner min in Equation 2.2 seems to be a typo. In Theorem 2.1, there is typo around the definition of $\xi$. Below thoerem 2. 1, the phrase "depend on the q-th moment" has 'q' undefined. Typo "is has a" in Theorem 3.1 is the the most insightful result, though simple to prove in hindsight. It would greatly help the readers if simple figures are used to explain this insightful result in the final manuscript.
(ROUGE-1: 0.36181, ROUGE-2: 0.07576, ROUGE-L: 0.35176) |

| | |
|---|---|
| *LED* | This paper presents a new way to measure the margin in the context of deep networks that removes the exponential dependency of depth in the corresponding generalization bounds. The reviewers agree that this is a valuable contribution both for theory and practices.
(ROUGE-1: 0.32836, ROUGE-2: 0.10050, ROUGE-L: 0.28856) |
| *PegasusX* | The paper presents a new generalization bound for deep neural networks and develops a regularizer which optimize quantities related to the bound and improve generalization error on competitive baselines. The paper is well-written and the main contributions come out clearly to non-specialists. However, the paper still requires rewriting to properly emphasize that it does not avoid exponential dependence on depth, but rather avoids an explicit dependence. The authors also need to discuss the computability of kappadv.
(ROUGE-1: 0.37975, ROUGE-2: 0.08511, ROUGE-L: 0.32911) |
| RAMMER | The paper shows that the all-layer margin has a clear and direct relationship with generalization for deep models. The paper also shows that this bound can be adapted to adversarial robustness of the deep model.
The paper is well-written and well-motivated. The reviewers found the paper to be a valuable contribution both for theory and practices.
(ROUGE-1: 0.29358, ROUGE-2: 0.04630, ROUGE-L: 0.27523) |

Table 11: Generated meta-reviews by fully-supervised PRIMERA, LED, PegasusX and RAMMER for random samples, and ROUGE scores measuring the quality of generated meta-reviews in comparison to the ground truth one.