# OpenReview forum: "Summarizing Multiple Documents with Conversational Structure for Meta-Review Generation"
_EMNLP/2023/Conference — EMNLP 2023 Findings_

### Official Review · Reviewer_h33m · 2023-08-01

**Soundness:** 4

**Excitement:**

4: Strong: This paper deepens the understanding of some phenomenon or lowers the barriers to an existing research direction.

**Paper Topic And Main Contributions:**

This paper constructed a new dataset PEERSUM for meta-review generation, which is different from other MDS datasets. This dataset includes inter-document relationships with an explicit hierarchical conversational structure and conflicting information, which is meaningful for paper review and abstractive summarization. In addition, it also proposes a baseline for this dataset, which considers multi-task framework and obtains an effectiveness results.

**Questions For The Authors:**

A. Explanation of the coefficients in Table 1
B. In relevant MDS papers, it is also mentioned that abstractive summarization needs to consider the inter-document relationship, why you said that other popular MDS datasets all lack inter-document relationships?
C. Figure 2 is a little confusing. How to explain the arcs of relationship-aware sparse attention? Also, the seven types of relationships in section 4.1, are not explained very clearly enough(why is the parent asymmetric relationship?).
D. For multi-task design, there is a lack of experiments related to analyzing the role of these auxiliary tasks.
E. Is there a specific design for conflict information in the dataset in the model？
F. Suggest hiding the author information of the paper from the case in the appendix



**Reasons To Accept:**

This paper proposes a new dataset for analyzing multi-document abstractive summarization, which is interesting and meaningful. The analysis and statistics of the dataset are relatively comprehensive and have high credibility.

**Reasons To Reject:**

1. The description of the key relationship-aware sparse attention is unclear, and the model diagram is quite difficult to understand for this part.
2. Since conflict is an important feature of this data, the proposed model does not clearly reflect the design of conflicts.
3. No analysis was conducted on the design of the four auxiliary tasks.

**Reproducibility:**

4: Could mostly reproduce the results, but there may be some variation because of sample variance or minor variations in their interpretation of the protocol or method.

**Reviewer Confidence:**

4: Quite sure. I tried to check the important points carefully. It's unlikely, though conceivable, that I missed something that should affect my ratings.

---

> ### Author Rebuttal · Authors · 2023-08-28
>
> Thanks for your great comments. Our responses regarding the questions are as follows.
>
> _A.“Explanation of the coefficients in Table 1.”_
>
> As there are no coefficients in Table 1, we assume that you mean the coefficients in Eq. 1. $\beta$ denotes learnable weights for different relationships in each attention head, as different relationships intuitively have different importance and effects on the meta-review generation process. We will clarify this in our revision.
>
> _B. “In relevant MDS papers, it is also mentioned that abstractive summarization needs to consider the inter-document relationship, why you said that other popular MDS datasets all lack inter-document relationships?”_
>
> That is because inter-document relationships are not explicitly provided in those datasets although inter-document relationships may exist in nature and should be considered in methodology, while we have them explicitly in PeerSum (the conversational structure and conflict indicators), which we believe is helpful for research that is trying to model and evaluate the recognition and resolution of inter-document relationships in summarization. We will clarify this in the revision.
>
> _C. “Figure 2 is a little confusing. How to explain the arcs of relationship-aware sparse attention? Also, the seven types of relationships in section 4.1, are not explained very clearly enough(why is the parent asymmetric relationship?).”_
>
> The arcs of relationship-aware sparse attention denote sparse connections among source documents based on the conversational structure.
>
> Relationships such as parent (ancestor-1) are asymmetric because we would have attention coming towards each node from its parent in the attention design for the relationship. As in the relationships, we use both the parent-to-child (ancestor-1) and child-to-parent (descendant-1) relationships in our design and they are intuitively of different importance and effect; therefore, we model them as asymmetric separately. We will clarify these relationships better by incorporating some diagrams in the revision.
>
> _D. “For multi-task design, there is a lack of experiments related to analyzing the role of these auxiliary tasks.”_
>
> Fair point. We will include ablation experiments to analyse the impact of each auxiliary task in the revision.
>
> _E. “Is there a specific design for conflict information in the dataset in the model?”_
>
> By providing explicit conversational structure to the model (via the sparse attention), we are hoping that it would help the model to recognise the conflict information in the dataset. But as our paper found, although it helps the model to **recognise** them (Table 8), it doesn’t do much to help **resolve** them, demonstrating the challenge and paving the path for future work.
>
> _F. “Suggest hiding the author information of the paper from the case in the appendix”_
>
> Thanks for your suggestions. We will do that.

---

### Official Review · Reviewer_ULo8 · 2023-08-05

**Soundness:** 3

**Excitement:**

4: Strong: This paper deepens the understanding of some phenomenon or lowers the barriers to an existing research direction.

**Paper Topic And Main Contributions:**

This paper proposes a novel dataset called PeerSum which includes abstracts and reviewer comments from ICLR and NeurIPS. The structures within documents and conversations make it different from other MDS problems. This paper also proposed a relationship-aware meta-review generation model with multi-task learning techniques. Their method achieves better performance than other baselines but still can be improved.

**Questions For The Authors:**

1. Why don't you use the same scheduling strategy (warm-up and scheduler) and decoding strategy (beam size and length penalty) for all baselines?

**Reasons To Accept:**

1. The meta-review generation task is different from other MDS tasks since there are rich structure information and metadata in the source documents.
2. Their method which uses sparse attention and multi-task learning outperforms other baseline models.
3. This paper offers some interesting observations, such as the model still struggles in solving conflicts between reviewers.
4. The source code and dataset are provided, so it can be easy to reproduce the experimental results.

**Reasons To Reject:**

1. The RAMMER Model is quite complex and it can be very difficult to balance between different tasks.
2. There are differences in the hyper-parameter settings from one baseline to another, so the experimental results may have some bias.

**Reproducibility:**

5: Could easily reproduce the results.

**Reviewer Confidence:**

3: Pretty sure, but there's a chance I missed something. Although I have a good feel for this area in general, I did not carefully check the paper's details, e.g., the math, experimental design, or novelty.

---

> ### Author Rebuttal · Authors · 2023-08-28
>
> Thanks for your great comments. Our responses regarding the issues and questions are as follows.
>
> _1. “The RAMMER Model is quite complex and it can be very difficult to balance between different tasks.”_
>
> Agree. As the meta-data (i.e., reviewer confidences, paper acceptance, document types, and review ratings) all intuitively have an effect on the meta-reviewers’ writing process of meta-reviews, we are trying to incorporate as much meta-data as possible to construct the baseline. We will include more ablation experiments in the revision to better understand the impact of each sub-task and see whether we need all of them. One last comment about training convergence: We observed that the overall performance is not very sensitive to the scaling hyper-parameters for the different sub-tasks, so RAMMER is not a difficult model to train in our practice.
>
> _2. "There are differences in the hyper-parameter settings from one baseline to another, so the experimental results may have some bias." and "Why don't you use the same scheduling strategy (warm-up and scheduler) and decoding strategy (beam size and length penalty) for all baselines?"_
>
> We used the hyperparameter configurations as recommended by the original authors for all baselines (which are optimal settings). We made this choice for two reasons: (1) we want to make sure each model is performing optimally; and (2) we found tuning them did not improve the performance much.

---

### Official Review · Reviewer_iSYN · 2023-08-07

**Soundness:** 3

**Excitement:**

3: Ambivalent: It has merits (e.g., it reports state-of-the-art results, the idea is nice), but there are key weaknesses (e.g., it describes incremental work), and it can significantly benefit from another round of revision. However, I won't object to accepting it if my co-reviewers champion it.

**Paper Topic And Main Contributions:**

This paper present PEERSUM, a data set for generating meta reviews for scientific paper and RAMMER, a Relationship aware multi-task meta review generator model that predicts meta features for a paper.

**Reasons To Accept:**

Code and dataset will be released for other users to work on top of the current work.
New area of Scientific paper meta review generation explored.
Strong experiment setup and detailed experiment analysis

**Reasons To Reject:**

Results are not strong, and work needs further iterations to show significant impact.
Need stronger examples on how meta review generation can be used on real world use cases.


**Reproducibility:**

3: Could reproduce the results with some difficulty. The settings of parameters are underspecified or subjectively determined; the training/evaluation data are not widely available.

**Reviewer Confidence:**

3: Pretty sure, but there's a chance I missed something. Although I have a good feel for this area in general, I did not carefully check the paper's details, e.g., the math, experimental design, or novelty.

---

> ### Author Rebuttal · Authors · 2023-08-28
>
> Thanks for your effort in reviewing the paper. Our responses regarding the issues you pointed out are as follows.
>
> _1. “Results are not strong, and work needs further iterations to show significant impact.”_
>
> The main contribution of our paper is actually the introduction of the novel multi-document summarization task/dataset with explicit conversational structures and conflicts among source documents and the insights it gave to the multi-document summarization community, e.g., the extent the meta-reviews are grounded in reviews and the difficulty in conflict handling. Understanding inter-document structures and relationships would be a critical capability of ideal multi-document summarization systems, but to the best of our knowledge, there were no multi-document summarization datasets explicitly providing these structures/relationships in the literature. Our work would pave the way for the research on modelling and evaluating the recognition and resolution of inter-document relationships in text summarization. The model we present constitutes only a baseline for the task, and our experimental results demonstrate the difficulty of the task and opportunities for future work.  We present these in both the Introduction and Conclusion sections.
>
> _2. “Need stronger examples on how meta review generation can be used on real world use cases.”_
>
> From an application perspective, (1) generating drafts of meta-reviews could serve to reduce the workload of meta-reviewers, as meta-reviewing is a highly time-consuming process for many conferences; (2) summarization for meta-review generation can be transferred to other domains which also feature intricate inter-document relationships, such as literary analysis and exegesis, etc.
>
> From a scientific research perspective, the task serves as a probe that allows us to understand how machines can reason, aggregate and summarise potentially conflicting opinions. We will clarify these motivations and add use cases in the revision.

---

### Meta-Review · Area_Chair_2axB · 2023-09-17

**Recommendation:** 3

**Metareview:**

The paper introduces PEERSUM, an innovative dataset for generating meta-reviews of scientific papers, and RAMMER, a multi-task meta-review generator model. PEERSUM includes features such as explicit inter-document relationships and conflicting information, enhancing the realism in multi-document summarization. RAMMER utilizes sparse attention and multi-task learning, showing promise compared to baseline models. The paper's strengths include a rigorous experimental setup, code and data sharing commitments, and the potential for advancing meta-review generation. However, reviewers unanimously call for stronger results, real-world use cases, clearer model explanations, and improved analysis of hyper-parameters and auxiliary tasks, indicating areas for enhancement in the paper's overall impact.

---

### Decision · Program_Chairs · 2023-10-07

**Decision:**

Accept-Findings

**Comment:**

The paper introduces PEERSUM, an innovative dataset for generating meta-reviews of scientific papers, and RAMMER, a multi-task meta-review generator model. PEERSUM includes features such as explicit inter-document relationships and conflicting information, enhancing the realism in multi-document summarization. RAMMER utilizes sparse attention and multi-task learning, showing promise compared to baseline models. The paper's strengths include a rigorous experimental setup, code and data sharing commitments, and the potential for advancing meta-review generation. However, reviewers unanimously call for stronger results, real-world use cases, clearer model explanations, and improved analysis of hyper-parameters and auxiliary tasks, indicating areas for enhancement in the paper's overall impact.